

# A non-negative feedback self-distillation method for salient object detection

Lei Chen[1], Tieyong Cao[1], Yunfei Zheng[1,2,3], Jibin Yang[1], Yang Wang[1], Yekui Wang[1] and Bo Zhang[4]

[1] The Army Engineering University of PLA, Nanjing, China
[2] The PLA Army Academy of Artillery and Air Defense, Hefei, China
[3] The Key Laboratory of Polarization Imaging Detection Technology, Hefei, China
[4] Institute of International Relations, National Defense University of Science and Technology, Nanjing, China

## ABSTRACT

Self-distillation methods utilize Kullback-Leibler divergence (KL) loss to transfer the knowledge from the network itself, which can improve the model performance without increasing computational resources and complexity. However, when applied to salient object detection (SOD), it is difficult to effectively transfer knowledge using KL. In order to improve SOD model performance without increasing computational resources, a non-negative feedback self-distillation method is proposed. Firstly, a virtual teacher self-distillation method is proposed to enhance the model generalization, which achieves good results in pixel-wise classification task but has less improvement in SOD. Secondly, to understand the behavior of the self-distillation loss, the gradient directions of KL and Cross Entropy (CE) loss are analyzed. It is found that KL can create inconsistent gradients with the opposite direction to CE in SOD. Finally, a non-negative feedback loss is proposed for SOD, which uses different ways to calculate the distillation loss of the foreground and background respectively, to ensure that the teacher network transfers only positive knowledge to the student. The experiments on five datasets show that the proposed self-distillation methods can effectively improve the performance of SOD models, and the average $F_\beta$ is increased by about 2.7% compared with the baseline network.

## INTRODUCTION

Salient object detection (SOD) aims to estimate the visual saliency region, and is an important computer vision task (*Ali et al., 2019*). In recent years, with the rapid development of deep neural networks, the performances of SOD have been greatly improved. However, high-performance SOD networks usually require large network structures and a large number of computing resources (*Chen et al., 2020*).

In order to solve the problem of too large network structure, *Hinton, Vinyals & Dean (2015)* proposed knowledge distillation to improve the performance of lightweight networks. Knowledge distillation uses the knowledge transferred from the teacher network to guild the student network training, which can improve the performance of lightweight student network. Traditional knowledge distillation methods need to train a large-scale

Corresponding author
Tieyong Cao, cty_ice@sina.com

teacher network with good performance in advance; then the lightweight student network can improve its performance by learning the knowledge transferred from the teacher network. However, the pre-training teacher network still has the complex network structure, so *Zhang et al. (2019a)* proposed self-distillation methods to solve this problem. Self-distillation methods do not require an independent teacher network, and improve the network performance by distilling the knowledge from the student network itself.

To solve SOD problem, many researchers (*Tang, Li & Zou, 2020*; *Zhang et al., 2019b*) apply knowledge distillation, but pay less attention to self-distillation. Current researches on self-distillation mainly focus on classification tasks such as image classification (*Li et al., 2020*) and semantic segmentation (*Ji et al., 2021*). In order to improve the performance of SOD without increasing the network size, we introduce self-distillation into SOD, and propose a non-negative feedback self-distillation method for SOD.

The output of classification task is the category probability distribution (*Xu et al., 2020*). The output of SOD is the category probability (*Pang et al., 2022*). As the outputs are different, the self-distillation method used for classification tasks may not be suitable for SOD.

In classification tasks, the knowledge distillation structure usually uses Cross Entropy (CE) loss and Kullback–Leibler divergence (KL) loss to guide the student network training (*Kim et al., 2021*). CE generates classification loss, which guides the student network to match the ground truth of training samples. KL produces distillation loss, which guides the student to match the prediction probability of the teacher network. KL divergence can effectively measure the similarity between two distributions, and drive the student network to imitate the performance of the teacher network. However, when KL divergence formula is directly applied to SOD as distillation loss function of the output layer, it is found that the signs of the distillation loss and its derivative will be opposite to CE loss. That is, the optimization direction is not consistent between KL loss and CE loss. The negative feedback will be generated and attenuate the performance of SOD network.

To solve this problem, we propose a new non-negative feedback self-distillation method. First of all, inspired by regularization thought (*Li et al., 2020*), we construct a pixel-wise virtual teacher model that is based on the ground truth. We find that the virtual teacher model can achieve good performance in classification tasks, but is not suitable for SOD. Then, we analyze KL divergence and find out the cause of the negative feedback. Finally, using the ideas of KL and Focal Loss (*Lin et al., 2017*), a non-negative feedback distillation loss is proposed, which calculates the distillation losses of foreground and background by different ways and is suitable for SOD. The proposed non-negative feedback distillation loss can drive the transferring of the teacher knowledge to the student network. The main contributions of this paper are as follows:

(1) The reason why KL divergence formula is not a suitable distillation loss for SOD is analyzed. The optimization directions of KL and CE are inconsistent, which affect the network training.

(2) An improved distillation loss with non-negative feedback is proposed, which modifies the formulation of KL to eliminate the negative feedback and holds the same optimization direction with CE.

(3) Experiments on five datasets show that the self-distillation architecture can be applied to SOD; Our self-distillation method improves by about 3.8% in the average $E$ compared with the baseline network. Compared with other self-distillation methods, our method can obtain the best experiment results; Our self-distillation method improves by about 2% in the average $F_\beta$ compared with the second-best method (BYOT).

## RELATED WORK

### Salient object detection

Traditional SOD methods mainly use color (*Cheng et al., 2014b*), boundary prior (*Zhu et al., 2014*), and sparsity (*Li, Sun & Yu, 2015*) to obtain the object salient map. These methods can obtain more detail information, but they are difficult to obtain high-quality semantic information. The recently developed deep learning methods can obtain both the detail and semantic information, so more and more scholars (*Wei et al., 2020*; *Chen et al., 2020*; *Zhao et al., 2020*; *Wei, Wang & Huang, 2020*; *Wu, Su & Huang, 2019*; *Mao et al., 2021*) use deep learning methods to solve SOD problem. *Wei et al. (2020)* divided the feature map into two parts: body map and detail map, and refined the two parts respectively. *Chen et al. (2020)* adopted feature interweaved aggregation, self-refined, head attention and global context flow modules to construct a global context-aware progressive aggregation network. *Zhao et al. (2020)* designed a gated dual branch structure. Collaborations between features from different layers were established to improve the distinguishability of the entire network. *Wei, Wang & Huang (2020)* mixed features from different layers by designing cross feature modules and cascaded feedback decoders, to generate better salient maps. *Wu, Su & Huang (2019)* discarded the features from shallow layers to improve the computing efficiency, and refined the features from deep layers to improve their representation ability. *Chen et al. (2020)* introduced initial prediction, side-output residential learning and top-down reverse attention to solve the complex architecture problem. *Mao et al. (2021)* used swin transformer structure to mix multi-layer features, and used attention mechanism to strengthen feature representation ability. However, these methods usually have large network structures and are difficult to be directly applied to the reality.

Recently, *Zhang et al. (2019b)* applied KD to SOD, they reduced the channels amount to construct the student network, and adopted multi-scale to transfer knowledge from the teacher to the student. Besides *Piao et al. (2020)* introduced cross-modal distillation on RGB-D based SOD, they distilled the depth information through an adaptive distiller. However, less attention is paid to self-distillation to solve SOD problem.

### Self-distillation

Ordinary knowledge distillation requires an additional teacher network to guide the student network learning. Self-distillation methods do not need an additional teacher network, and the student network learn the knowledge from itself. Self-distillation methods mainly construct auxiliary branches to transfer the knowledge to the student network. *Ji et al. (2021)* used the auxiliary teacher network to refine the soft label and feature map knowledge, which can better preserve the local information. *Hou et al. (2019)* let the shallow feature learn the deep feature expression, so as to strengthen the overall feature expression ability.

*Zhang et al. (2019a)* divided the original network into several shallow networks according to the characteristics of the network structure, and distilled the separated shallow networks respectively. *Li & Chen (2020)* used consistency strategy to match the knowledge that is extracted from auxiliary branches.

*Li et al. (2020)* proved that self-distillation is a special label smoothing regularization method. *Yun et al. (2020)* used the predicted distribution of training samples as distillation knowledge. Therefore, *Li et al. (2020)* and *Yun et al. (2020)* considered that self-distillation is a special regularization method.

At the same time, *Xu & Liu (2019)* and *Lee, Hwang & Shin (2020)* considered that self-distillation is a special data augmentation method. *Xu & Liu (2019)* used the distorted versions of training samples as distillation knowledge. *Lee, Hwang & Shin (2020)* used the transformations of training samples as distillation knowledge.

The regularization method does not require complex branch structures and matching strategies, and can further simplify the scale of network parameters. We use pixel-wise regularized distribution to construct a self-distillation framework.

## METHOD

In this section, we firstly introduce a virtual teacher self-distillation architecture; secondly analyze the distillation loss calculated by KL in classification task, and compare it with CE; thirdly analyze the distillation loss calculated by KL formula in SOD; finally propose our non-negative feedback distillation loss.

As the principles of multi classification tasks and binary classification tasks are the same, we take binary classification task as an example for analysis. Given a training sample $X1 = x_i, i = 1, \ldots, W \times H$, $x_i$ is the $i$-th pixel in the sample, $W$ and $H$ are the width and height of the sample, $Y1 = y_i, i = 1, \ldots, W \times H$ is the corresponding ground truth. In order to facilitate the description, $y_i = 1$ means the pixel belongs to the object category; $y_i = 0$ means the pixel belongs to the background category. The sample output is $p^s$ in the student network and $p^t$ in the teacher network.

### Virtual teacher self-distillation architecture

*Li et al. (2020)* used the regularized category probability as virtual knowledge to construct a teacher model, and manually set the output probability of the teacher network. Based on this, we used the regularized probability distribution to construct a pixel-wise virtual teacher model. Unlike *Li et al. (2020)*, who addressed multi-classification problems, we have extended their approach to tackle pixel-wise segmentation problem. The self-distillation structure is shown in Fig. 1. The output probability of the teacher network is as follows:

$$p^t(a_i) = \begin{cases} \mu, & if \ a_i = y_i \\ \dfrac{(1-\mu)}{(K-1)}, & if \ a_i \neq y_i \end{cases} \tag{1}$$

where $K$ is the total number of categories, $K$ is 2 in SOD; $y_i$ is the correct label; and $a_i$ is the prediction label; $\mu$ is the predict probability of correct pixel classification. Usually, $\mu \geq 0.9$ is set (*Li et al., 2020*), to ensure that the probability of correct pixel classification

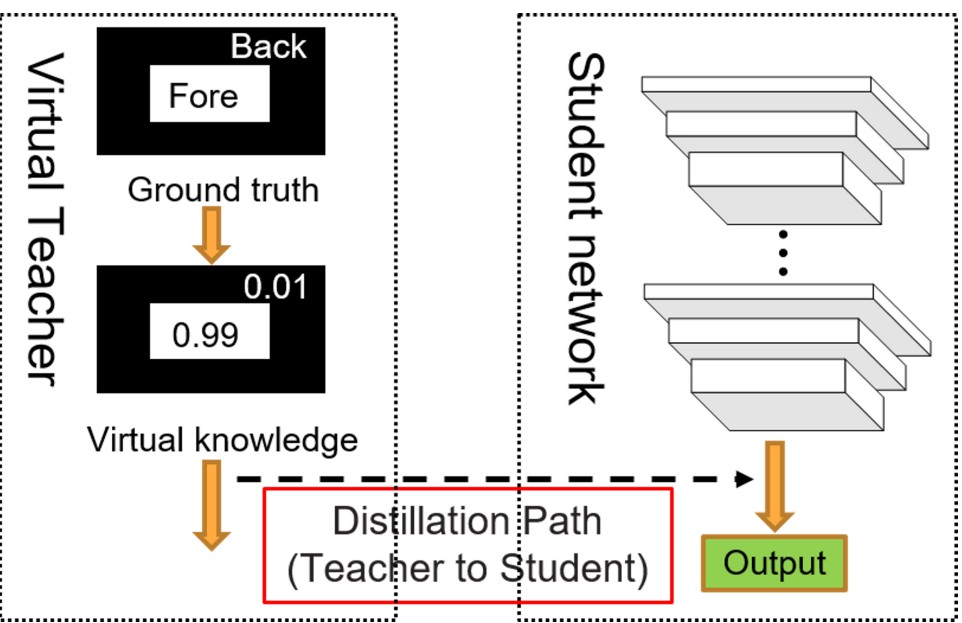

**Figure 1** Virtual teacher self-distillation architecture.

is far greater than wrong classification; we set $\mu = 0.99$. When the pixel is the labeled foreground, the output probability of the virtual teacher is 0.99; when the pixel is the labeled background, the output probability of the virtual teacher is 0.01. If the value of $\mu$ is small, the predict probability of the student will be greater than the teacher in easy and well classified pixels. In these pixels, the teacher cannot transfer knowledge to the student, and will bring negative effect to the network training. Therefore, we set $\mu$ as 0.99, and *Li et al. (2020)* proved it in experiments.

We use the backbone network of F3Net (*Wei, Wang & Huang, 2020*) as the student network (the right box in Fig. 1), and build a self-distillation learning framework on this basis. The virtual teacher provides correct knowledge to the student network, and guild the student network to optimize. Different from hard label learning which hopes the output of the teacher and student are the same, this self-distillation method hopes that the output distribution of the student fits the teacher output distribution. Virtual teacher self-distillation method provides more distribution information while making the student results the same as the teacher.

For a training set with $N$ samples, $D = \{(X_j, Y_j) | 1 \leq j \leq N\}$, where $X_j \in R^{H*W*3}$ is the $j$-th sample, $H$ and $W$ are the height and width of the sample, $Y_j$ is the corresponding ground truth. $W_M = \{W_m | m = 1, \ldots, L\}$ represents the learnable weight matrix of a $L$-layer neural network. The training goal of the neural network is to learn a mapping function $f(W_m; X) : X \rightarrow Y$. The most common training method is Empirical Risk Minimization (*Wang, 2021*). The neural network parameter $W_m$ can be adjusted by optimizing the following functions.

$$\underset{W_m}{\text{Arg min}} \, L_{mt}(W_m; D), \qquad\qquad (2)$$

where $L_{mt}$ is the total loss of all training samples. In the self-distillation framework for SOD, the loss function is determined by pixel-wise CE and KL loss.

$$L_{mt} = L_{KL}\left(p^s, p^t\right) + L_{CE}\left(W_m; D\right), \tag{3}$$

$$L_{KL}\left(p^s, p^t\right) = \frac{1}{N * W * H} \sum_{j=1}^{N} \sum_{i=1}^{W*H} p_{ij}^s \log \frac{p_{ij}^s}{p_{ij}^t}, \tag{4}$$

$$L_{CE}\left(W_m; D\right) = -\frac{1}{N * W * H} \sum_{j=1}^{N} \sum_{i=1}^{W*H} \left[y_{ij}\log p_{ij}^s + \left(1 - y_{ij}\right)\log\left(1 - p_{ij}^s\right)\right], \tag{5}$$

where $p_{ij}^s$ is the prediction probability of the student network for $i$-th pixel in $j$-th sample; $p_{ij}^t$ is the prediction probability of the corresponding teacher network; $y_{ij}$ is the corresponding annotated label, which is 1 when the pixel belongs to the foreground and is 0 when the pixel belongs to the background.

It is found that this virtual teacher self-distillation architecture can improve the model performance for classification tasks, but achieve little improvement for SOD. Especially on easily classified datasets with significant differences between foreground and background, the model performance can hardly be improved. To apply the virtual teacher self-distillation method to SOD, we analyze KL loss and propose a new loss to replace KL loss.

## Distillation loss analysis in classification task

In classification task, self-distillation structure usually uses Cross Entropy loss ($L_{CE}$) and Kullback–Leibler divergence loss ($L_{KL}$) to guide the student network training (*Kim et al., 2021*). The sample loss $L$ is calculated as follows:

$$L = L_{KL}\left(p^s \| p^t\right) + L_{CE}\left(p^s, Y\right). \tag{6}$$

In binary classification task, Cross Entropy (CE) is Binary Cross Entropy (*Li et al., 2019*). *Hossain, Betts & Paplinski (2021)* proved that when the pixel belongs to the object, CE loss is positive and the loss derivative is negative; when the pixel belongs to the background, CE loss is positive, and the loss derivative is positive. The formulas of CE loss and loss derivative are shown in Table 1, the loss curves are shown in Fig. 2, and the loss derivative curves are shown in Fig. 3. The optimization direction of a good distillation loss should be consistent with CE. If the optimization direction is not consistent, the negative feedback will be generated. The negative feedback affects the network optimization, and leads to the poor performance.

KL divergence $L_{KL}$ is calculated by Eq. (7) (*Li et al., 2020*). As the temperature parameter does not affect the optimization direction of the loss and loss derivative, the influence of temperature parameter is not considered.

$$L_{KL}\left(p^s \| p^t\right) = \frac{1}{W * H} \sum_{i=1}^{W \times H} \vec{p_i^s} \log \frac{\vec{p_i^s}}{\vec{p_i^t}}, \tag{7}$$

where $\vec{p_i^s}$ is the output of the student network, which denotes a $1 \times k$ dimensional array. $K$ is the number of categories, it is 2 in binary classification task. $\vec{p_i^t}$ is the output of the

**Table 1  The formulas of CE loss and loss derivative.**

| | Object pixel ($y_i=1$) | Background pixel ($y_i=0$) |
| --- | --- | --- |
| $L_{CE}$ | $-\log p^s$ | $-\log(1-p^s)$ |
| $\frac{dL_{CE}}{dp^s}$ | $-1/p^s$ | $1/(1-p^s)$ |

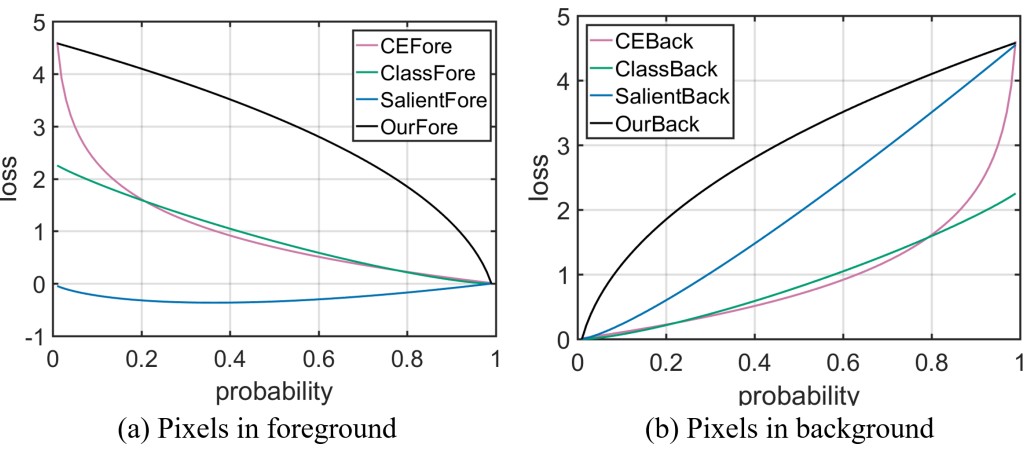

(a) Pixels in foreground        (b) Pixels in background

**Figure 2  Relationship between pixels' predicted probability and loss.** The horizontal coordinate represents pixel predicted probability, the vertical coordinate represents pixel loss value. CEFore and CEBack represent Cross Entropy loss; ClassFore and ClassBack represent distillation loss in binary classification task; SalientFore and SalientBack represent distillation loss in SOD; OurFore and OurBack represent non-negative feedback distillation loss.

corresponding teacher network.

$$\vec{p}_i^s = [p1_i^s, p2_i^s, \dots, pk_i^s], \tag{8}$$

where $p1_i^s, p2_i^s, \dots, pk_i^s$ are the output probability predicted by the softmax function, and their summation is 1. In binary classification task, the output of the student network can be expressed as follows:

$$\vec{p}_i^s = [p_i^s, 1-p_i^s], \tag{9}$$

where $p_i^s$ is the probability that the student network predicts the pixel $i$ as the object. The KL loss of the pixel $i$ is calculated as follows:

$$L_{KL}^i \left( \vec{p}_i^s \big\| \vec{p}_i^t \right) = \frac{1}{2}\left[ p_i^s \log \frac{p_i^s}{p_i^t} + \left(1-p_i^s\right) \log \frac{1-p_i^s}{1-p_i^t} \right], \tag{10}$$

where $p_i^t$ is the probability that the teacher network predicts the pixel $i$ as the object, and the value of $p_i^t$ is (0,1).

The derivation of $L_{KL}^i$ is as follows:

$$\frac{dL_{KL}^i}{dp_i^s} = \frac{1}{2}\left[ \log \frac{p_i^s}{p_i^t} + \log \frac{1-p_i^t}{1-p_i^s} \right]. \tag{11}$$

In self-distillation framework, most works construct auxiliary teacher branches to generate refined knowledge, or adopt deep-level knowledge to guild the shallow-level

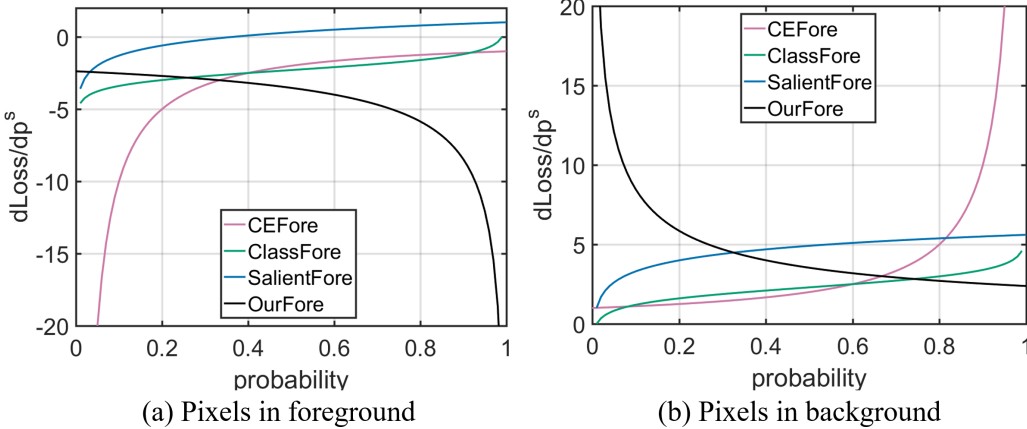

**Figure 3** **Relationship between pixels' predicted probability and loss derivative.** The horizontal coordinate represents pixel predicted probability, the vertical coordinate represents pixel loss derivative value. CEFore and CEBack represent Cross Entropy loss; ClassFore and ClassBack represent distillation loss in binary classification task; SalientFore and SalientBack represent distillation loss in SOD; OurFore and OurBack represent non-negative feedback distillation loss.

training. The purposes of these works are to construct the teacher which performance is better than the student network (*Hinton, Vinyals & Dean, 2015*). When the pixel belongs to the object, the output probability of the teacher network is greater than the student network, that is $p_i^t \geq p_i^s$. So $(\mathrm{d}L_{KL}^i/\mathrm{d}p_i^s) \leq 0$, $L_{KL}^i$ is a monotone decreasing function in the range of values. $p_i^t \geq p_i^s$, the maximum value of $p_i^s$ is $p_i^t . L_{KL}^i(p_i^s = p_i^t) = 0$, the minimum value of $L_{KL}^i$ is 0. So, $L_{KL}^i$ is greater than 0. Therefore, when the pixel belongs to the object, the distillation loss calculated by KL divergence is greater than 0 and the loss derivative is less than 0. In the object, the optimization directions of the distillation loss and loss derivative are consistent with CE.

When the pixel belongs to the background, the output probability of the teacher network is less than the student network, that is $p_i^t \leq p_i^s$. So $(\mathrm{d}L_{KL}^i/\mathrm{d}p_i^s) \geq 0$, $L_{KL}^i$ is a monotone increasing function in the range of values. $p_i^t \leq p_i^s$, the minimum value of $p_i^s$ is $p_i^t . L_{KL}^i(p_i^s = p_i^t) = 0$, the minimum value of $L_{KL}^i$ is 0. So, $L_{KL}^i$ is greater than 0. Therefore, when the pixel belongs to the background, the distillation loss calculated by KL divergence is greater than 0 and the loss derivative is greater than 0. In the background, the optimization directions of the distillation loss and loss derivative are consistent with CE.

For a more intuitive presentation, we assume that the teacher network output probability $p_i^t$ is 0.99 in the object pixel and 0.01 in the background pixel. At this time, the distillation loss and loss derivative formulas are shown in Table 2, the loss curves are shown in Fig. 2, and the loss derivative curves are shown in Fig. 3.

From Figs. 2 and 3, it can be seen that the signs of distillation loss and loss derivative value are consistent with CE. As the optimization direction of the distillation loss which is calculated by KL divergence is consistent with CE, distillation loss can better guide the student network training in binary classification task.

**Table 2  In binary classification task, the formulas of distillation loss and loss derivative.**

|  | object pixel ($y_i=1$) | background pixel ($y_i=0$) |
|---|---|---|
| $L_{KL}^i$ | $\frac{1}{2}\left[p_i^s\log\frac{p_i^s}{p_i^t}+(1-p_i^s)\log\frac{1-p_i^s}{1-p_i^t}\right]$ | $\frac{1}{2}\left[p_i^s\log\frac{p_i^s}{p_i^t}+(1-p_i^s)\log\frac{1-p_i^s}{1-p_i^t}\right]$ |
| $\frac{dL_{KL}^i}{dp_i^s}$ | $\frac{1}{2}\left[\log\frac{p_i^s}{p_i^t}+\log\frac{1-p_i^t}{1-p_i^s}\right]$ | $\frac{1}{2}\left[\log\frac{p_i^s}{p_i^t}+\log\frac{1-p_i^t}{1-p_i^s}\right]$ |

## Distillation loss analysis in salient object detection

Similar to the classification task, we also use CE loss ($L_{CE}$) and distillation loss ($L_{KD}$) to guide the student network training in SOD. Then the sample loss $L$ is calculated as follows:

$$L = L_{KD}(p^s, p^t) + L_{CE}(p^s, Y). \tag{12}$$

In SOD, CE is also the binary CE (*Wei, Wang & Huang, 2020*). That is, when the pixel belongs to the foreground, CE loss is positive and the loss derivative is negative. When the pixel belongs to the background, CE loss is positive, and the loss derivative is positive.

When KL formula is used to calculate the distillation loss in SOD, the distillation loss is calculated as follows:

$$L_{KD}(p^s, p^t) = L_{KL}(p^s\|p^t) = \frac{1}{W*H}\sum_{i=1}^{W\times H}\left[y_i p_i^s\log\frac{p_i^s}{p_i^t}+(1-y_i)p_i^s\log\frac{p_i^s}{p_i^t}\right], \tag{13}$$

where $y_i = 0$ means the pixel belongs to the background; $y_i = 1$ means the pixel belongs to the foreground; $p_i^s$ and $p_i^t$ are the outputs of the student and teacher networks, which are *1*-dimensional values. The KL loss of the pixel $i$ is calculated as follows:

$$L_{KL}^i(p^s\|p^t) = p_i^s\log\frac{p_i^s}{p_i^t}. \tag{14}$$

The derivation of $L_{KL}^i$ is as follows:

$$\frac{dL_{KL}^i}{dp_i^s} = \log\frac{p_i^s}{p_i^t}+1. \tag{15}$$

In self-distillation framework, most works construct auxiliary teacher branches to generate refined knowledge, or adopt deep-level knowledge to guild the shallow-level training. The purposes of these works are to construct the teacher which performance is better than the student network (*Hinton, Vinyals & Dean, 2015*). When the pixel belongs to the foreground, the output probability of the teacher network is greater than the student network, $p_i^t \geq p_i^s.L_{KL}^i(p^s\|p^t)$ is less than 0. At this time, the sign of distillation loss is inconsistent with CE. When $0 < (p_i^s/p_i^t) \leq (1/e)$, that is $(dL_{KL}^i/dp_i^s) \leq 0$. At this time, the sign of loss derivative is consistent with CE. When $(1/e) < (p_i^s/p_i^t) \leq 1$, that is $(dL_{KL}^i/dp_i^s) \geq 0$. At this time, the sign of loss derivative is inconsistent with CE. Therefore, in the foreground, the optimization direction of distillation loss will be inconsistent with CE, resulting in negative feedback, which will affect the student network performance.

When the pixel belongs to the background, the output probability of the teacher network is less than the student network, $p_i^t \leq p_i^s.L_{KL}^i(p^s\|p^t)$ is greater than 0; $(dL_{KL}^i/dp_i^s)$ is greater

**Table 3  In SOD, the formulas of distillation loss and loss derivative.**

| | foreground pixel ($y_i$=1) | background pixel ($y_i$=0) |
|---|---|---|
| $L_{KL}^i$ | $p_i^s \log \frac{p_i^s}{p_i^t}$ | $p_i^s \log \frac{p_i^s}{p_i^t}$ |
| $\frac{dL_{KL}^i}{dp_i^s}$ | $\log \frac{p_i^s}{p_i^t} + 1$ | $\log \frac{p_i^s}{p_i^t} + 1$ |

than 1. At this time, the optimization directions of distillation loss and loss derivative are consistent with CE. For a more intuitive presentation, we assume that the teacher network output probability $p_i^t$ is 0.99 in the foreground pixel and 0.01 in the background pixel. The distillation loss and loss derivative formulas are shown in Table 3, the loss curves are shown in Fig. 2, and the loss derivative curves are shown in Fig. 3.

Combining Figs. 2 and 3 and the above analysis, it can be seen that when KL formula is used to calculate the distillation loss in SOD, the optimization direction of the distillation loss and loss derivative are inconsistent with CE in the foreground. The performance improvement of the student network is limited. Therefore, it is defective to directly use KL formula to calculate the distillation loss in SOD.

## Non-negative feedback distillation loss (NKL)

In order to transfer the knowledge, the optimization direction of distillation loss should be consistent with CE. In SOD, when the pixel belongs to the foreground, the loss is greater than 0, and the loss derivative is less than 0; when the pixel belongs to the background, the loss is greater than 0, and the loss derivative is greater than 0.

Inspired by KL, CE and Focal loss (*Lin et al., 2017*), we propose a non-negative feedback distillation loss, which uses different formulas to respectively calculate foreground and background distillation loss. The loss is calculated as follows:

$$L_{NKL}\left(p^s, p^t\right) = \frac{1}{W * H} \sum_{i=1}^{W \times H} \left[ y_i \left(1 - p_i^s\right)^\alpha \log \frac{1 - p_i^s}{1 - p_i^t} + \left(1 - y_i\right)\left(p_i^s\right)^\alpha \log \frac{p_i^s}{p_i^t} \right]. \quad (16)$$

where $y_i = 0$ means the pixel belongs to the background; $y_i = 1$ means the pixel belongs to the foreground; $\alpha$ is a hyperparameter that is greater than 0 and less than 1, which is determined by experiments and is selected as 0.3 here.

When the pixel belongs to the foreground, the distillation loss and loss derivative of the pixel $i$ are calculated as follows:

$$L_{NKL}^{i+}\left(p^s, p^t\right) = \left(1 - p_i^s\right)^\alpha \log \frac{1 - p_i^s}{1 - p_i^t}, \quad (17)$$

$$\frac{dL_{KL}^{i+}}{dp_i^s} = \left[\alpha\left(1 - p_i^s\right)^{\alpha-1}\right] \log \frac{1 - p_i^t}{1 - p_i^s} - \left(1 - p_i^s\right)^{\alpha-1}. \quad (18)$$

At this time, $1 \geq p_i^t \geq p_i^s \geq 0$. Therefore, $L_{NKL}^{i+}\left(p^s, p^t\right)$ is greater than 0, $(dL_{KL}^{i+}/dp_i^s)$ is less than 0. Therefore, when the pixel belongs to the foreground, our distillation loss is greater

than 0 and the loss derivative is less than 0. In the foreground, the optimization directions of our distillation loss and loss derivative are consistent with CE.

When the pixel belongs to the background, the distillation loss and loss derivative of the pixel $i$ are calculated as follows:

$$lL_{NKL}^{i-}\left(p^s,p^t\right) = \left(p_i^s\right)^{\alpha}\log\frac{p_i^s}{p_i^t}, \tag{19}$$

$$\frac{\mathrm{d}L_{KL}^{i-}}{\mathrm{d}p_i^s} = \alpha(p_i^s)^{\alpha-1}\log\frac{p_i^s}{p_i^t} + \left(p_i^s\right)^{\alpha-1}. \tag{20}$$

At this time, $0 \leq p_i^t \leq p_i^s \leq 1$. Therefore, $L_{NKL}^{i-}\left(p^s,p^t\right)$ is greater than 0, $(\mathrm{d}L_{KL}^{i-}/\mathrm{d}p_i^s)$ is greater than 0. Therefore, when the pixel belongs to the background, our distillation loss is greater than 0 and the loss derivative is greater than 0. In the background, the optimization directions of our distillation loss and loss derivative are consistent with CE. For a more intuitive presentation, we assume that the teacher network output probability $p_i^t$ is 0.99 in the foreground pixel and 0.01 in the background pixel. The distillation loss and loss derivative formulas are shown in Table 4, the loss curves are shown in Fig. 2, and the loss derivative curves are shown in Fig. 3.

As the optimization direction of our non-feedback distillation loss is consistent with CE in the foreground and background, it can better guide the student network training in SOD.

So, in virtual teacher self-distillation architecture, we replace Eq. (4) as Eq. (21).

$$L_{NKL}\left(p^s,p^t\right) = \frac{1}{N*W*H}\sum_{j=1}^{N}\sum_{i=1}^{W*H}\left[y_{ij}\left(1-p_{ij}^s\right)^{\alpha}\log\frac{1-p_{ij}^s}{1-p_{ij}^t} + \left(1-y_{ij}\right)\left(p_{ij}^s\right)^{\alpha}\log\frac{p_{ij}^s}{p_{ij}^t}\right]. \tag{21}$$

In order to verify the universality of our method, we also apply our non-negative feedback distillation loss to other self-distillation frameworks. Through experiments in 'Comparison with recent self-distillation methods' we find that our non-negative feedback loss function can work in other self-distillation methods and work better in our virtual teacher method. The main reason is that the teacher network performance may be worse than the student in easy and well classified pixels in other self-distillation methods. In non-negative feedback loss function, the worse performance teacher cannot guild the student training, which leads to the limited improvement to the student network. While in the virtual teacher method, the teacher performance is always better than the student.

# EXPERIMENT

## Experimental configurations
### Model

We use the backbone network of F3Net which is based on Resnet-50 (*He et al., 2016*) as the student network. For the convenient analysis, we remove the branches of F3Net, only use its backbone network as the baseline network. The loss function is CE. We use this model as the baseline network. During training, the maximum learning rate is 0.005, and warmup (*Priya et al., 2017*) and linear decay strategies are used to dynamically adjust the

**Table 4   The formulas of non-negative feedback distillation loss and loss derivative.**

| | Foreground pixel ($y_i=1$) | Background pixel ($y_i=0$) |
|---|---|---|
| $L_{KL}^i$ | $(1-p_i^s)^\alpha \log \frac{1-p_i^s}{1-p_i^t}$ | $(p_i^s)^\alpha \log \frac{p_i^s}{p_i^t}$ |
| $\frac{dL_{KL}^i}{dp_i^s}$ | $[\alpha(1-p_i^s)^{\alpha-1}]\log\frac{1-p_i^t}{1-p_i^s} - (1-p_i^s)^{\alpha-1}$ | $\alpha(p_i^s)^{\alpha-1}\log\frac{p_i^s}{p_i^t} + (p_i^s)^{\alpha-1}$ |

learning rate. The training strategy is Stochastic Gradient Descent (SGD). The momentum and weight decay are set to 0.9 and 0.0005 respectively. In the experiments, the batchsize is 32, the maximum epoch is 32, and all image sizes are set to 352*352.

### Datasets

We conducted experiments on five challenging datasets with salient or camouflaged objects. The five datasets are COD (*Fan et al., 2020*), DUT-O (*Yang et al., 2013*), THUR (*Cheng et al., 2014a*), PASCAL-S (*Li et al., 2014*) and HKU-IS (*Li & Yu, 2015*). COD is the dataset with natural camouflaged objects, including 6066 natural images and corresponding pixel-wise annotation images. DUT-O, THUR, PASCAL-S and HKU-IS are salient object datasets, which respectively contain 4,447, 5,168, 850, 1,447 images and corresponding annotation images. COD is divided into training and testing sets by the default setting; DUT-O and THUR are divided into training and testing sets by the proportion of 0.6 and 0.4. PASCAL-S and HKU-IS are divided into training and testing sets by the proportion of 0.8 and 0.2.

### Metrics

We use $F_\beta$ Measure ($F$) (*Fan et al., 2020*), the mean absolute error (MAE) (*Yang et al., 2021*), E-measure ($E$) (*Kang & Kang, 2021*) and precision–recall (PR) curve (*Xian et al., 2022*) to evaluate the network performance.

$F$ is the weighted mean of precision and recall. The calculation formula is as follows:

$$F_\beta = \frac{(1+\beta)*\text{precision}*\text{recall}}{\beta*precision+\text{recall}}, \tag{22}$$

where $\beta$ is the weight, usually is set to 0.3; precision focuses on the accuracy of the object detection; Recall focuses on the integrity of the object detection.

MAE is calculated as follows:

$$\text{MAE} = \frac{1}{H\times W}\sum_{i=1}^{H}\sum_{i=1}^{W}\left|P(i,j)-G(i,j)\right|, \tag{23}$$

where $H$ and $W$ is the height and width of the sample, $P$ is the prediction result of the network, and $G$ is the ground truth. $F$ and $E$ are larger, MAE is smaller, the network performance is better.

### Hyperparameter selection

We discuss the selection of $\alpha$ in Eq. (16). We select different $\alpha$ to test the model performance. Through experiments, we choose $\alpha = 0.3$, when the model achieves the best performance. The result is shown in Table 5.

**Table 5** The results of different hyperparameters (%).

| $\alpha$ | COD | | DUT-O | |
|---|---|---|---|---|
| | F | MAE | F | MAE |
| 0.2 | 67.08 | 5.7 | 83.11 | 4.4 |
| 0.3 | **67.58** | **5.1** | **83.89** | **4.1** |
| 0.5 | 66.00 | 5.1 | 83.15 | 4.2 |
| 0.8 | 65.79 | 5.6 | 82.08 | 4.6 |
| 1 | 66.09 | 5.5 | 82.71 | 4.5 |
| 1.5 | 65.83 | 6.0 | 52.56 | 4.8 |
| 2 | 65.16 | 6.2 | 82.25 | 5.0 |

**Notes.**
The bold values mean the best results.

The following conclusions are drawn from Table 5. (1) When $\alpha$ takes different values, the model performance all can be improved. (2) When $\alpha$ is greater than a certain value, the model performance begins to decline. This shows that the influence of output probability is not the bigger, the model performance is better. When the influences of misclassified pixels are too great, the model may be affected just by these pixels. The model only obtains optimal result in these pixels.

## Comparison with different one-dimensional distance metric methods

In SOD, the output is the category probability which is one-dimension, the aim of self-distillation is that the student network produces the same distribution with the teacher. Root Mean Square Error (RMSE), Mean Absolute Error (MAE), Mean Squared Error (MSE), Cosine Similarity (CS) are the common one-dimensional distance metric functions and are widely used as the loss functions. Therefore, we use these functions and KL formula (KL) as distillation loss functions, and compare them with our method (NKL). In the experiment, virtual teacher self-distillation architecture remained unchanged and only changed the distillation loss function. The experiment results are shown in Table 6.

The following conclusions can be drawn from Table 6. (1) Not all one-dimensional distance metric methods can be used as distillation loss function. From the mean performance over five datasets, when RMSE is used as the distillation loss, the network performance after distillation is poorer than before distillation. From the evaluating indicator MAE, when KL formula is used as distillation loss, the network performance after distillation is poorer than before distillation. (2) Our method (NKL) can transfer the knowledge well. Our method can achieve the best detection results on all five datasets. Especially, from the mean E, our method improves 2.2% compare with the second-best method (KL); from the mean MAE, our method reduces 1.6% compare with KL. These prove that our method is effective.

## Comparison with recent self-distillation methods

First, we use the backbone network of F3Net as the baseline and take this as the student network. Then, FR (*Ji et al., 2021*), SA (*Hou et al., 2019*), BYOT (*Zhang et al., 2019a*) and DHM (*Li & Chen, 2020*) self-distillation methods are introduced into the baseline. Finally,

**Table 6  The results of different distillation loss functions (%).**

| Methods | | Baseline | KL | RMSE | MAE | MSE | CS | NKL |
|---|---|---|---|---|---|---|---|---|
| COD | F | 64.16 | 67.08 | 63.96 | 64.31 | 64.36 | 59.90 | **67.58** |
| | MAE | 6.3 | 6.2 | 6.2 | 6.2 | 6.1 | 6.4 | **5.1** |
| | E | 73.8 | 76.3 | 73.6 | 74.0 | 74.0 | 75.0 | **80.8** |
| DUT-O | F | 81.25 | 83.11 | 81.10 | 80.72 | 81.38 | 79.81 | **83.89** |
| | MAE | 5.1 | 5.6 | 5.1 | 5.2 | 5.1 | 4.9 | **4.1** |
| | E | 85.6 | 86.7 | 85.6 | 85.1 | 85.4 | 87.8 | **89.9** |
| THUR | F | 86.74 | 88.26 | 86.60 | 87.01 | 87.09 | 86.98 | **89.33** |
| | MAE | 3.5 | 3.9 | 3.5 | 3.5 | 3.4 | 3.1 | **2.6** |
| | E | 90.3 | 91.7 | 90.2 | 90.5 | 90.5 | 92.1 | **93.3** |
| PASCAL-S | F | 82.18 | **84.52** | 81.86 | 82.74 | 82.35 | 80.84 | 84.29 |
| | MAE | 8.9 | 9.6 | 9.1 | 8.8 | 8.7 | 8.9 | **7.0** |
| | E | 81.4 | 83.3 | 81.2 | 81.5 | 81.0 | 81.6 | **83.2** |
| HKU-IS | F | 87.94 | 90.18 | 88.25 | 88.36 | 88.77 | 88.09 | **90.57** |
| | MAE | 4.9 | 5.0 | 4.8 | 4.8 | 4.6 | 4.3 | **3.5** |
| | E | 91.5 | 92.7 | 91.9 | 92.0 | 92.5 | 93.0 | **94.4** |
| MEAN | F | 80.45 | 82.63 | 80.35 | 80.63 | 80.79 | 79.12 | **83.13** |
| | MAE | 5.74 | 6.06 | 5.74 | 5.70 | 5.58 | 5.52 | **4.46** |
| | E | 84.52 | 86.14 | 84.50 | 84.62 | 84.68 | 85.90 | **88.32** |

**Notes.**
The bold values mean the best results.

KL formula and our non-negative feedback loss function (NKL) are respectively used as distillation loss to train the network. The experiment results are shown in Table 7.

SA directly uses pixel-wise attention features in the backbone network as distillation knowledge. FR uses Bi-directional Feature Pyramid Network (BiFPN) to generate pixel-wise knowledge. Therefore, SA and FR can be directly applied to SOD. BYOT and DHM cannot directly generate pixel-wise knowledge. Therefore, we modify them to ensure that they can be applied to the self-distillation framework for SOD. We mainly make the following two modifications. (1) We adjust the step size of the first convolution layer in the bottleneck module from 2 to 1. This operation is to maintain the spatial size of the feature in the bottleneck module of the auxiliary branch. (2) We change the full connected layer of the auxiliary branch to the convolution layer. This operation is to generate pixel-wise knowledge that can be transferred to the backbone network.

From Table 7, we draw the following conclusions. (1) Self-distillation methods are also suitable for SOD. The results in the table show that self-distillation methods can improve the network performance. Compared with the baseline, our method improves the average *F* by nearly 2% and the average *E* by nearly 3.8%. (2) Our virtual teacher model is better than other self-distillation methods. In the five datasets, our virtual teacher model can achieve the best detection results. Especial, from the mean *F*, our method improves nearly 2% compare with the second-best method (BYOT); from the mean *E*, our method improves nearly 3% compare with the second-best method (BYOT). (3) Our non-negative feedback loss function (NKL) achieves better results than KL formula in different self-distillation

**Table 7 The results of recent self-distillation methods (%).**

| Methods | | Baseline | BYOT | | DHM | | SA | | FR | | OUR | |
|---|---|---|---|---|---|---|---|---|---|---|---|---|
| | | | KL | NKL | KL | NKL | KL | NKL | KL | NKL | KL | NKL |
| COD | F | 64.16 | **65.16** | 64.81 | 64.12 | **64.61** | 64.32 | **64.73** | **64.34** | 63.70 | 67.08 | **67.58** |
| | MAE | 6.3 | **6.0** | 6.0 | 6.2 | **6.1** | 6.1 | **6.1** | **6.1** | 6.2 | 6.2 | **5.1** |
| | E | 73.8 | **74.7** | 74.8 | 74.0 | **74.1** | 73.8 | **73.9** | **74.1** | 73.6 | 76.3 | **80.8** |
| DUT-O | F | 81.25 | 81.36 | **82.17** | 80.95 | **81.25** | 80.93 | **81.22** | 80.93 | **81.66** | 83.11 | **83.89** |
| | MAE | 5.1 | 5.0 | **5.0** | 5.2 | **5.2** | 5.2 | **5.2** | 5.1 | **5.0** | 5.6 | **4.1** |
| | E | 85.6 | 85.8 | **86.3** | 85.2 | **85.4** | 85.2 | **85.3** | 85.8 | **85.8** | 86.7 | **89.9** |
| THUR | F | 86.74 | 87.43 | **87.66** | 86.91 | **87.02** | **87.17** | 87.13 | 87.08 | **87.26** | 88.26 | **89.33** |
| | MAE | 3.5 | 3.4 | **3.4** | 3.5 | **3.5** | **3.5** | 3.5 | 3.4 | **3.4** | 3.9 | **2.6** |
| | E | 90.3 | 91.0 | **91.2** | 90.6 | **90.7** | **90.9** | 90.5 | 90.6 | **90.9** | 91.7 | **93.3** |
| PASCAL-S | F | 82.18 | 82.29 | **82.33** | 82.27 | **82.32** | 81.89 | **82.08** | 82.27 | **83.11** | 84.52 | 84.29 |
| | MAE | 8.9 | 8.9 | **8.7** | 9.0 | **8.8** | 9.0 | **8.9** | 8.9 | **8.9** | 9.6 | **7.0** |
| | E | 81.4 | 81.4 | **81.5** | 81.3 | **81.8** | 81.1 | **81.7** | 80.7 | **81.9** | 83.3 | 83.2 |
| HKU-IS | F | 87.94 | **88.66** | 88.47 | 87.70 | **88.23** | 88.64 | **88.66** | 88.42 | **88.89** | 90.18 | **90.57** |
| | MAE | 4.9 | **4.6** | 4.6 | 4.8 | **4.8** | 4.8 | **4.7** | 4.7 | **4.6** | 5.0 | **3.5** |
| | E | 91.5 | **92.3** | 92.2 | 91.5 | **91.9** | 92.2 | **92.5** | 91.9 | **92.4** | 92.7 | **94.4** |
| MEAN | F | 80.45 | 80.98 | **81.01** | 80.39 | **80.69** | 80.59 | **80.77** | 80.61 | **80.93** | 82.63 | **83.13** |
| | MAE | 5.74 | 5.58 | **5.54** | 5.74 | **5.68** | 5.72 | **5.68** | 5.64 | **5.62** | 6.06 | **4.46** |
| | E | 84.52 | 85.04 | **85.2** | 84.52 | **84.78** | 84.64 | **84.78** | 84.62 | **84.92** | 86.14 | **88.32** |

**Notes.**
The bold values mean the best results.

methods in SOD. In DHM, NKL can achieve better detection results in five datasets. Among other methods, NKL can also achieve better results in at least three datasets. And KL is mainly better than our method in COD. As the foreground and background are similar in camouflaged images, the prediction result of the teacher network may be worse than that of the student network in other self-distillation methods. In NKL, the worse performance teacher cannot guild the student training, which limits the improvement of the network performance. (4) NKL can work better in our virtual teacher method than other self-distillation methods. From the mean $E$, NKL improves nearly 2% compare with KL in virtual teacher, but improves nearly 0.3% in other self-distillation methods. The main reason is that the teacher network performance may be worse than the student in easy and well classified pixels in other self-distillation methods. In NKL, the worse performance teacher cannot guild the student training, which leads to the limited improvement to the student network.

## Comparison with recent SOD methods

We compared with five SOD methods, namely LDF (*Wei et al., 2020*), GCP (*Chen et al., 2020*), RAS (*Chen et al., 2020*), GateNet (*Zhao et al., 2020*), CPD (*Wu, Su & Huang, 2019*). All methods are used their default learning rate, momentum, weight decay and maximum epochs.

Table 8 quantitatively shows the detection results of different methods. It can be seen that our method can achieve good detection results on five datasets, and can achieve

**Table 8** The results of recent SOD methods (%).

| Methods | | LDF | GCP | RAS | GateNet | CPD | OUR |
|---|---|---|---|---|---|---|---|
| COD | F | 67.93 | 67.88 | **69.54** | 65.42 | 57.75 | 67.58 |
| | MAE | 5.3 | 5.8 | **4.6** | 5.9 | 7.7 | 5.1 |
| | E | 79.3 | 75.2 | **83.8** | 74.4 | 71.6 | 80.8 |
| DUT-O | F | 84.27 | **84.16** | 84.28 | 82.81 | 82.63 | 83.89 |
| | MAE | 4.1 | 4.2 | 4.1 | 4.5 | 5.1 | **4.1** |
| | E | 89.0 | 88.6 | **90.0** | 87.0 | 87.5 | 89.9 |
| THUR | F | 89.29 | 88.85 | 88.03 | 87.80 | 87.18 | **89.33** |
| | MAE | 2.6 | 2.7 | 2.8 | 3.2 | 3.1 | **2.6** |
| | E | 93.1 | 92.3 | 92.6 | 91.7 | 91.8 | **93.3** |
| PASCAL-S | F | 84.11 | **85.16** | 84.88 | 84.98 | 83.27 | 84.29 |
| | MAE | 7.2 | 7.7 | 7.5 | 7.4 | 8.7 | **7.0** |
| | E | 82.9 | 82.8 | 83.1 | **84.2** | 82.0 | 83.2 |
| HKU-IS | F | 89.24 | 90.01 | 87.05 | 89.39 | 89.11 | **90.57** |
| | MAE | 4.0 | 4.2 | 4.5 | 4.1 | 4.2 | **3.5** |
| | E | 93.3 | 93.4 | 91.4 | 93.1 | 93.0 | **94.4** |
| MEAN | F | 82.97 | **83.21** | 82.75 | 82.08 | 79.99 | 83.13 |
| | MAE | 4.64 | 4.92 | 4.7 | 5.02 | 5.76 | **4.46** |
| | E | 87.52 | 86.46 | 88.18 | 86.08 | 85.18 | **88.32** |

**Notes.**
The bold values mean the best results.

the best detection results on THUR and HKU-IS. From the mean performance over five datasets, our method also achieves the best detection results. From the mean $E$, our method improves nearly 3% compare with CPD.

Figure 4 shows the precision–recall curves of different methods. It can be seen that our curves are higher than other methods in COD, THUR and HKU-IS, which prove that our method can achieve good performance.

Table 9 shows the detection efficiency of different methods. We compare the model efficiency from the model parameter size and detection speed. It can be seen that our method has the smallest parameter scale and the fastest detection speed, when our model performance is similar to other methods. From the size of params, our method reduces near 100M compare with GateNet.

## CONCLUSIONS

Self-distillation has been proven to improve the performance of the lightweight network, and is widely used in computer vision tasks. However, when self-distillation is applied to SOD, the common distillation loss function (KL divergence) will generate negative feedback. In order to solve this problem, a non-negative feedback distillation loss is proposed. The experiment results show that our method can improve the network performance. As the advantages of self-distillation, more and more tasks will make use of self-distillation in the future. Our method further expands the application scope of self-distillation, and provides a new attempt to adopt self-distillation for new tasks.

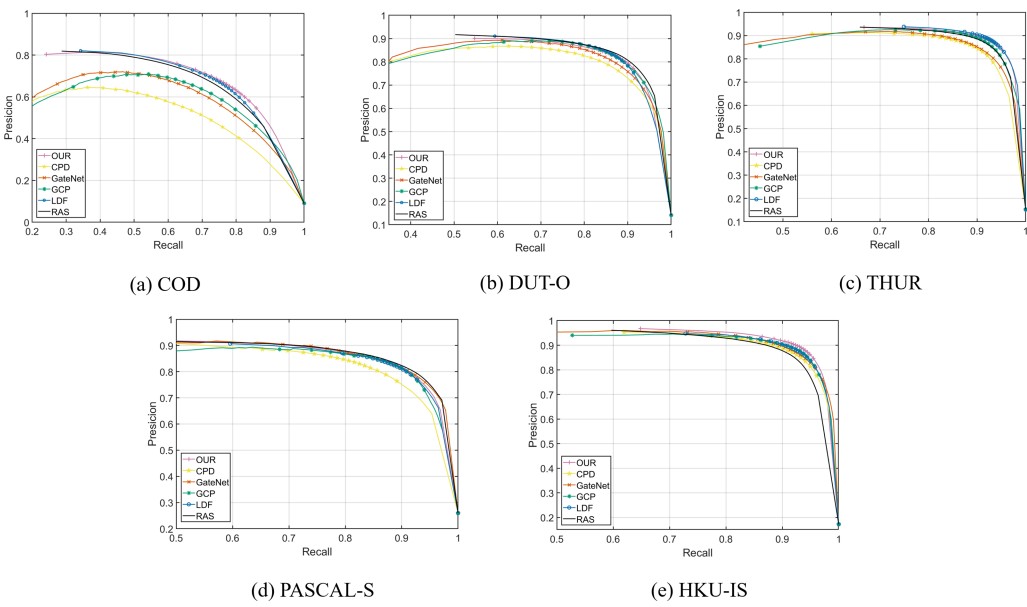

(a) COD  (b) DUT-O  (c) THUR

(d) PASCAL-S  (e) HKU-IS

**Figure 4** PR curve of different methods.

**Table 9** The efficiency comparison with recent SOD methods.

| Metric | LDF | GCP | RAS | GateNet | CPD | OURS |
|--------|-----|-----|-----|---------|-----|------|
| #Params (MB) | 25.15 | 67.06 | 24.70 | 128.63 | 47.85 | **23.79** |
| Speed (FPS) | 47.56 | 50.52 | 50.34 | 41.53 | 38.72 | **74.42** |

**Notes.**
The bold values mean the best results.

### Funding

This work was supported by the Natural Science Foundation of China (61801512, 62071484), the Natural Science Foundation of Jiangsu Province (BK20180080) and the University of National Defense Science and Technology 2021 School Scientific Research project (ZK21-43). The funders had no role in study design, data collection and analysis, decision to publish, or preparation of the manuscript.

### Grant Disclosures

The following grant information was disclosed by the authors:
Natural Science Foundation of China: 61801512, 62071484.
Natural Science Foundation of Jiangsu Province: BK20180080.
University of National Defense Science and Technology 2021: ZK21-43.

### Competing Interests

The authors declare there are no competing interests.

## Author Contributions

- Lei Chen conceived and designed the experiments, performed the experiments, analyzed the data, performed the computation work, prepared figures and/or tables, authored or reviewed drafts of the article, and approved the final draft.
- Tieyong Cao conceived and designed the experiments, analyzed the data, authored or reviewed drafts of the article, and approved the final draft.
- Yunfei Zheng conceived and designed the experiments, analyzed the data, performed the computation work, authored or reviewed drafts of the article, and approved the final draft.
- Jibin Yang analyzed the data, authored or reviewed drafts of the article, and approved the final draft.
- Yang Wang performed the experiments, prepared figures and/or tables, authored or reviewed drafts of the article, and approved the final draft.
- Yekui Wang analyzed the data, prepared figures and/or tables, authored or reviewed drafts of the article, and approved the final draft.
- Bo Zhang analyzed the data, prepared figures and/or tables, authored or reviewed drafts of the article, and approved the final draft.

## Data Availability

The code is available at GitHub and Zenodo: https://github.com/chenlei1011/Self-distillation-NKL-code.git.

chenlei1011. (2023). chenlei1011/Self-distillation-NKL-code: First release of my Self-distillation-NKL-code (v1.0.0). Zenodo. https://doi.org/10.5281/zenodo.7894479.

The data is available at:

– PASCAL-S dataset: Georgia Tech, Yin Li et al. (2014) The secrets of salient object segmentation. https://doi.org/10.1109/CVPR.2014.43.

– COD dataset: https://dengpingfan.github.io/pages/COD.html. Media Computing Lab, College of Computer Science, Nankai University, Deng-Ping Fan et al. (2020) Camouflaged object detection. https://doi.org/10.1109/CVPR42600.2020.00285.

– HKU-IS dataset: https://i.cs.hku.hk/~gbli/deep_saliency.html. The University of Hong Kong, Guanbin Li et al. (2015) Visual saliency based on multiscale deep features. https://doi.org/10.1109/CVPR.2015.7299184.

– DUT-OMRON dataset: http://saliencydetection.net/dut-omron/. Dalian University of Technology, Chuan Yang et al. (2013) Saliency detection via graph-based manifold ranking. https://doi.org/10.1109/CVPR.2013.407.

– THUR dataset: https://mmcheng.net/code-data/. Media Computing Lab, College of Computer Science, Nankai University, Ming-Ming Cheng et al. (2014) SalientShape: group saliency in image collections. https://doi.org/10.1007/s00371-013-0867-4.

## Supplemental Information

Supplemental information for this article can be found online at http://dx.doi.org/10.7717/peerj-cs.1435#supplemental-information.

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
