# Peer review of "A non-negative feedback self-distillation method for salient object detection"

_PeerJ Computer Science, doi:10.7717/peerj-cs.1435_

## Round 0.1 · original submission · Major Revisions

This is an interesting idea, but it needs significant modification. Both reviewers raised many detailed issues and asked the authors to address these concerns thoroughly.

Reviewer 1 ·

Basic reporting

• In general, an abstract should specify the content of the article completely. For instance a format like 2–3 lines of Introduction to the concepts used followed by 2–3 lines of proposed method and objectives in short, and finalizing the abstract with the method used, its measurement in improvement of performance towards the existing method –when explained in this pattern will give an impact of how the article will take the reader to travel through the concept. Lines 17- 27 could be modified based on this aspect to make a budding researcher/ reader understand the concept of the article.
• Lines 24- 26 should definitely use some metrics or evaluation data to strongly say that their proposed system excels the lightweight SOD.

Experimental design

The experiment is performed as shown in section 4. Experiments. This corresponds to the code of reference 13. Mentioned in lines 405,406.
But the code and data used in the data availability statement links to a github page https://github.com/chenlei1011/Self-distillation-NKL-code which is a code of a published article @inproceedings{F3Net,
title = {F3Net: Fusion, Feedback and Focus for Salient Object Detection},
author = {Jun Wei, Shuhui Wang, Qingming Huang},
booktitle = {AAAI Conference on Artificial Intelligence (AAAI)},
year = {2020}
}
The authors found in the published article are not the same authors of this article, so this backbone architecture of the published article can be cited and the original code used to obtain the result and the experimental values should be provided. Lines 268-273 should be implemented in the code on the datasets as specified in lines 274-276.

Validity of the findings

• In line 291 it is specified alpha(α)is calculated based on equation 16, but is the author sure about it. The definition of alpha value is not specified in the equation.
• Findings defined in tables and figures are to be discussed regarding their improvement or their failures in appropriate places of discussion.

Additional comments

• Line 43, 44 should specify the reference articles where the author found the current research works on self-distillation. So that the purpose of introducing self distillation in the proposed system would be supported.

• Lines 46, 47 should be supported by references related to it.

• Lines 54, 55,57,58,64,65- check on the sentence formation
• The third contribution 3 in Lines 68,69 should be rewritten to say that the author contribution is towards proving the proposed method excels the existing methods in terms of the parameters(mention the parameters).

• In most of the sentences “ some scholars/ researchers”[lines 33,39,89,101, 103….] are used- kindly do not generalize , instead specify the references and the methods.
• “Some” is the word most often used as some cases, some pixels… these generalizations should be avoided.

• References used in related works would be different from that of the references used in introduction. Like 4,5,7,8…

• Line 113,121,134,152,153 doesnot convey the meaning the author strives to. Kindly check to it.

• Article usage and grammatical errors are to be concentrated.
• Line 140 needs reference citation for Empirical Risk Minimization

• Provide justification for line 215.
• Clarity in line 217 and 218 should be done. The condition preceeds or succeeds the statement is not clear due to the use of fullstop(.)

• Line 294 , line 307 and line 328 a close bracket ‘)’ is included. It is preferable to use them as bullets.

• In some citation Fig.* is used whereas in some citations Figure is used. Kindly abide by the format.

• References should be framed as per template of the journal.

Annotated reviews are not available for download in order to protect the identity of reviewers who chose to remain anonymous.
Cite this review as

Reviewer 2 ·

Basic reporting

The idea is interesting, but it needs to be significantly modified.
The English expression of the whole paper needs to be improved.
The order of the graphs and text in the paper is confusing. Please reorder them.
Line 47, “In some cases, the self-distillation method used for classification tasks may not be suitable for SOD.”. Please carefully describe in what circumstances .
Line 72, “Traditional SOD methods...” need to explain or require the suitable reference for better readability of work.
Section 2, related work seems to be incomplete, as it doesn’t cover all state-of-the-art methods, that need to be discussed. Use papers from year 2020 to 2023 specially.
Lines 270 and 273, “the maximum learning rate is 0.005” and “the maximum epoch is 20” need to be discussed, especially “epoch ”. Similarly, in Section 4.4 the epoch of each model is set to 20. Does the model converge?

Experimental design

no comment

Validity of the findings

no comment

Additional comments

no comment

Cite this review as

---

## Round 0.2 · accepted · Accept

The required comments were rectified and were included in the manuscript.

Reviewer 1 ·

Basic reporting

The required comments were rectified and were included in the manuscript.

Experimental design

The experimental design had few drawbacks which were not clearly stated in the version 1 of the article and the revised manuscript and code seems satisfactory.

Validity of the findings

The findings were supported with relevant code and supporting graphs. Thanks for it

Cite this review as

Reviewer 2 ·

Basic reporting

I have no more comments.

Experimental design

I have no more comments.

Validity of the findings

I have no more comments.

Additional comments

I have no more comments.

Cite this review as